# Peptide Drug Conjugates and Their Role in Cancer Therapy

**DOI:** 10.3390/ijms24010829

**Published:** 2023-01-03

**Authors:** Ethan Heh, Jesse Allen, Fabiola Ramirez, Daniel Lovasz, Lorena Fernandez, Tanis Hogg, Hannah Riva, Nathan Holland, Jessica Chacon

**Affiliations:** Paul L. Foster School of Medicine, Department of Medical Education, Texas Tech University Health Sciences Center, El Paso, TX 79905, USA

**Keywords:** peptide-drug conjugates, bioconjugates, payload, linker, carrier

## Abstract

Drug conjugates have become a significant focus of research in the field of targeted medicine for cancer treatments. Peptide-drug conjugates (PDCs), a subset of drug conjugates, are composed of carrier peptides ranging from 5 to 30 amino acid residues, toxic payloads, and linkers that connect the payload to the peptide. PDCs are further broken down into cell-penetrating peptides (CPPs) and cell-targeting peptides (CTPs), each having their own differences in the delivery of cytotoxic payloads. Generally, PDCs as compared to other drug conjugates—like antibody-drug conjugates (ADCs)—have advantages in tumor penetration, ease of synthesis and cost, and reduced off-target effects. Further, as compared to traditional cancer treatments (e.g., chemotherapy and radiation), PDCs have higher specificity for the target cancer with generally less toxic side effects in smaller doses. However, PDCs can have disadvantages such as poor stability and rapid renal clearance due to their smaller size and limited oral bioavailability due to digestion of its peptide structure. Some of these challenges can be overcome with modifications, and despite drawbacks, the intrinsic small size of PDCs with high target specificity still makes them an attractive area of research for cancer treatments.

## 1. Introduction

Cancer is a significant cause of death worldwide, with breast and lung cancer having the highest prevalence among women and lung and prostate cancer having the highest prevalence among men [1]. Other leading cancers, according to the world health organization (WHO), include cancers of the colon and rectum, stomach, liver, cervix, and esophagus. Cancer disproportionally affects minority communities within the United States [2,3]. Additionally, the WHO estimates that cancer accounts for 8.97 million deaths worldwide, making cancer the second highest cause of mortality worldwide behind cardiovascular disease [1]. The financial implication of cancer is also severe, with an estimated personal healthcare spending of an estimated 155.5 billion dollars in 2013 in the United States [4].

Historically, the first-line therapy for cancer was surgical excision of a primary tumor [5]. During the 20th century, radiation and chemotherapeutic such as aminopterin, doxorubicin, and cisplatin became available. However, traditional chemotherapy and radiation broadly target rapidly dividing cells, including non-cancerous cells such as hair follicles or enterocytes. Although the standard of care, these methods indiscriminately target cancerous and non-cancerous rapidly proliferating cells, which accounts for the side effects associated with classical chemotherapeutic or radiation treatment, such as hair loss and gastrointestinal upset. Despite advances in oncopharmacology such as immunotherapy, off-target cytotoxicity remains a chief concern, and efforts to mitigate these effects by increasing the targeting specificity of new chemotherapeutic agents.

One approach to reducing these side effects is to develop therapeutics that directly target cells of interest; this was first achieved in the 21st century using monoclonal antibodies to target specific antigens overexpressed or mutated in cancer cells. More selective methods of targeting cancer have developed, such as using chemotherapeutic agents linked to antibodies to deliver therapy to cancer cells. These drugs are known as antibody-drug conjugates (ADCs). Despite their advantages, especially compared to standard chemotherapeutics, ADCs have several drawbacks, including antibody aggregation, premature ADC dissociation, high rates of drug clearance, ineffective tumor penetration, and bystander effect leading to off-target cytotoxicity [6,7]. To combat some limitations of ADCs, smaller cancer-targeting chemotherapeutics were developed using a peptide-linked mechanism, termed peptide drug conjugates (PDCs). PDCs, much like ADCs, are constructed of 3 parts: (1) peptide carrier, (2) cytotoxic payload, and (3) linker [8]. Overall, the peptide carrier promotes tumor targeting, the payload (therapeutic) facilitates anticancer biological effects, and the linker connects the peptide carrier to the payload (Figure 1).

The key differences between PDCs and ADCs are generally (1) Immunogenicity, (2) Penetration and (3) Elimination. 

Immunogenicity: In the context of this review, immunogenicity can be defined as the unwanted ability for a drug or molecule to induce an immune response due to the body recognizing the infused drug as foreign and eliciting a response against it. PDCs generally are considered to have low immunogenicity characteristics since peptides have low intrinsic immunogenicity [9] ADCs on the other hand can exhibit increased risk for immunogenicity compared to PDC. This is due to the various domains in the ADC, such as the epitopes, linker and cytotoxic agent that can result in the development of anti-drug antibodies (ADAs) [10]. ADAs could potentially inactivate the drug, causing decreased effectiveness and targeting and overall suboptimal exposure [10].

Penetration: The weight of the PDC molecule is small (~2–20 kDa), allowing the PDC to penetrate the tumor stroma and enter the tumor cells [9]. In comparison, ADCs have a large molecular weight (∼160 kDa) that results in the limitation of transport through solid tumor cell surface [10]. In a study conducted by Chalouni and Doll [11] the internalization rate of an anti-CD30 mAb based ADC was determined using flow cytometry. It was demonstrated that 60% of the initial level of surface bound cAC10 antibody remained after 20 h [11].

Elimination: PDCs generally have a short half-life and are rapidly eliminated by the kidneys [9]. In addition, due to their small size, PDCs have the ability to reach tumor sites unattainable by larger molecules, such as ADCs. ADCs have a longer half-life than PDCs and are non-specifically taken up by the liver, resulting in potential dose-limiting toxicity to the liver [10].

The strategy and innovation of PDCs utilize the biological activities and potential of small-molecule peptides to improve the efficacy of treatments [12]. Since the first point of interaction between a drug or delivery vehicle and a cell is the exterior surface, any abnormally expressed receptors in cancerous tissues provide a focal point for targeted delivery [7]. Tumor cells often have receptors for unique peptide sequences [13,14,15]. Since the first point of interaction between a drug or delivery vehicle and a cell is the exterior surface, any abnormally expressed receptors in cancerous tissues provide a focal point for targeted delivery [12]. 

There have been various attempts at synthesizing PDCs for cancer treatment. In 2018 ^177^Lu-DOTA-TATE (DOTA-TATE) (Figure 2) was approved by the Food and Drug Administration (FDA) and is considered a first-in-class PDC. DOTA-TATE is used to treat somatostatin receptor-positive gastroenteropancreatic neuroendocrine tumors (GEP-NETs) [16]. Overall, this results in a sufficient amount of drug delivered to the cancer site while minimizing contact with healthy tissue. Herein, we present a detailed review of recent advances in the field of PDCs, including a summary of current clinical trials and the challenges that PDCs face in the era of precision medicine.

## 2. Peptide Drug Conjugates

Although ADCs are clinically established for cancer therapy, PDCs are gaining recognition as a new cancer treatment method by increasing targeted drug delivery with improved efficacy and reduced side effects. PDCs utilize a smaller molecular composition than other marketed anticancer drugs (such as ADCs), contributing to PDC biochemical stability, cell membrane penetration, and overall efficacy [17]. PDCs can be modified to optimize binding affinities and physicochemical properties to ensure proper binding and cleavage [9]. PDCs are classified as cell-penetrating peptides (CPPs) or cell-targeting peptides (CTPs).

### 2.1. Peptides for Specific Organ Targeting

Directed targeting of specific organs has been considered a crucial step in limiting side effects associated with traditional anticancer therapy. Use of peptides to direct organ specific targeting has emerged as a distinct possibility. Currently there are two main ways to target a peptide 1) rely on natural protein sequences such as vascular endothelial growth factor (VEGF) [18] or somatostatin [19]. Alternatively libraries of peptides can be tested via phage display technique [20]. However, these techniques often yield peptides that can target tumor microenvironments but are poorly directed to specific organs. Likewise certain organs are more easily targeted than others for example N-acetylgalactosamine (GalNAc) can be used to easily target the lungs in adenocarcinoma [21,22]. However, some organs prove more difficult to effectively target when a strong physical barrier is in place as is the case with pancreatic cancers characterized by strong desmoplasia creating a mechanical barrier around the tumor cells [23] or cancers of the brain that necessitate crossing the blood–brain barrier (BBB). That said, developments are underway to utilize phage-derived shuttle peptides which can select against BBB endocytic machinery and used in engineering novel PDCs for brain cancers [24].

### 2.2. Cell-Penetrating Peptides

The cell membrane provides a physiological barrier that limits the transportation of various molecules, such as macromolecules, proteins, and nucleic acids, across the plasma membrane. However, the cell membrane can also limit drug penetration. Therefore, it is imperative to develop drugs that can cross the cell membrane of cancer cells to induce destruction.

Cell-penetrating peptides (CPPs) can transport drug payloads through cell membranes using specific amino acid sequences ranging from 5 to 30 residues. CPPs provide an effective method for transporting cell-impermeable compounds or drugs to reach their intracellular targets [25]. Various mechanisms of action have been proposed regarding how CPPs penetrate the cell. Two generally accepted mechanisms are (1) direct penetration of the plasma membrane and (2) endocytosis (Figure 3). Direct penetration occurs when positively charged CPPs interact with negatively charged membrane components, destabilizing the membrane and forming a pore [25,26]. Moreover, clathrin-mediated endocytosis and macropinocytosis have been observed to take up CPPs [27]. However, more research needs to be done to elucidate the exact mechanism of cellular entry [28].

Due to the ability of CPPs to enter most cells they come into contact with, their therapeutic effects are limited to intra-tumoral injection. However, some treatments have been developed to target lymphatic metastasis via intravenous injection using CPPs modified with nanoparticles; in a study by Hu et al., modifying nanoparticles with CPPs suppressed tumor growth rate by 1.4-fold and showed a 63.3% inhibition rate of lymph metastasis in lung cancer [29]. Other advancements have been for specific tumor targeting by activatable cell-penetrating peptides and transducible agents. Coupling shielding polyanions create activatable CPPs (ACPPs) to the peptide with target-specific cleavable linkers [27]. For example, in a study by Cheng et al., the shielding group of 2,3-dimethyl maleic anhydride (DMA) was used to inhibit the CPP at physiological pH. However, at a tumor extracellular pH of 6.8, DMA is hydrolyzed to activate the CPP to sequester the drug inside cancer cells [30]. Transducible agents delivered via intraperitoneal injection use functional domains to modulate the type of tissues CPPs are active against to increase tumor specificity. One notable example is the creation of oxygen-dependent degradation (ODD) domains by fusing hypoxia-inducible factor-1α to β-galactosidase, which helps it target hypoxic tumor cells. This domain is combined with the HIV-TAT protein to reduce tumor growth without causing toxic side effects expected from delivering active caspase-3 [12].

### 2.3. Cell-Targeting Peptides

Cell-targeting peptides (CTPs) range from 3–14 amino acids long and utilize receptors that are overexpressed on cancer cell surfaces to target the delivery of the drug [25]. Depending on the targeted receptor, CTPs can cause a localized build-up of the drug around the tumor or induce endocytosis upon CTP binding (Figure 3) [31,32]. CTPs exhibit similar characteristics to monoclonal antibodies (mAbs) by binding with high affinity to their respective receptor. However, unlike mAbs, CTPs can penetrate tumors better due to their small size [9].

A limitation of CTPs includes the dependency on the expression of a specific receptor to have an effect. Techniques such as phage display can determine a peptide sequence that will specifically bind cancer cells and mitigate the shortcoming mentioned earlier [31]. In a study by Rasmussen et al., phage display was utilized to identify peptide sequences with a 1000-fold or higher binding efficiency and selectivity specific to human colorectal cancer cells [33]. Further, cyclization and multimerization can increase an affinity for the selected receptors. Cyclization forces the peptide into a constrained ring conformation, increasing the resistance to proteases and degradation. Multimerization joins two or more monomers together to improve local concentrations of CTPs, resulting in higher probabilities of peptide-receptor interactions [31]. CTPs can be combined with a CPP to translocate cargo molecules into cancer cells more efficiently. Bolhassani et al. found that delivery of a DNA alkylating agent, chlorambucil, with CREKA (CTP) conjugated to *p*VEC (CPP) (*p*VEC amino acid sequence LLIILRRRIRKQAHAHSK) was more suitable for transportation and led to significantly higher cancer killing than chlorambucil alone [34].

## 3. Linker Region

The linker region of a PDC connects the peptide to the drug, as demonstrated in Figure 1. The linker is vital in preventing off-target or peripheral drug release and delivering as much medication to the target (cancer cell) as possible. Linkers play this role by attenuating drug activity while connecting the drug and peptide. Once the PDC reaches the target, the linker is cleaved, and the drug is released in its fully active state. Linkers must be stable while in circulation but preferentially cleaved at the target site, thus ensuring the maximum possible amount of the drug dose reaches the target. Multiple types of linkers utilize different cancer or intracellular characteristics to achieve this preferential drug release. The main classes of linkers are non-cleavable, pH-sensitive, enzyme sensitive, and redox-sensitive.

### 3.1. Non-Cleavable Linkers

By their name, non-cleavable linkers seem paradoxical, as an essential function of linkers is to be cleaved at the right time. However, with any linker type, there will be inevitable amounts of cleavage in circulation and interstitially, resulting in premature drug release and side effects [35]. Non-cleavable linkers receive their name because they have a lower chance of being cleaved off-target than other linkers. The drug is typically made into an active state with some or all of the linker still attached to the drug [35,36]. This defense against cleaving is due to the stability of bonds in non-cleavable linkers that connect the peptide and drug to the linker. As exemplified by the chemical structure of ^177^Lu-DOTA-TATE (Figure 2) non-cleavable linkers usually consist of amide or ester bonds, which are unlikely to break extracellularly [25,37]. However, even with better stability, non-cleavable linkers are not as preferred as cleavable linkers [36].

### 3.2. pH-Sensitive Linkers

Linkers in the pH-sensitive group connect the drug and peptide to the linker through bonds that will break in a more acidic environment compared to plasma and interstitial space. The most typical pH-sensitive linker contains a hydrazone moiety [36] (Figure 4). These linkers will be stable in circulation at the plasma pH of 7.4 but then break when exposed to acidic environments around the tumor at a pH of 6.5–6.9 or inside endosomes and lysosomes with pH of 5.5–6.2 and 4.5–5.0, respectively [35,36]. Tumors will have this local acidic pH due to increased metabolic demands that create acidic byproducts [38].

### 3.3. Enzyme-Sensitive Linkers

Enzyme-sensitive linkers will exploit the increased amounts of specific enzymes expressed by tumors or inside lysosomes [35]. These linkers can contain ester and amide bonds similar to the non-cleavable linkers. Still, instead of having only carbon bonds with the ester/amide bonds, enzyme-sensitive linkers include specific amino acid sequences that allow enzyme targeting [35] such as a valine-citrulline sequence which is recognized by lysosomal cathepsin B (Figure 4). Ester and amide bonds will be targeted by esterases and proteases/amidases, which are common within tumors and inside lysosomes. Still, more specific bonds can be utilized depending on the enzymes expressed by cancer [35].

### 3.4. Redox-Sensitive Linkers

Redox-sensitive linkers exploit intracellular antioxidants like glutathione, which typically have intracellular concentrations 1000 times greater than plasma (from 15 mM intracellularly to 15 µM extracellularly) [39]. Glutathione can be even more concentrated in tumors due to local hypoxic conditions from poor blood flow that promotes increased production of reactive oxygen species [39,40]. Redox-sensitive linkers, therefore, attach to the drug through disulfide bonds (Figure 4), which will be broken by glutathione once inside the cell, but be stable outside the cell where glutathione’s concentration is significantly less.

## 4. Payload

Like most cancer treatments, the goal is to target cancerous cells or tumors and limit on-target off-tumor effects. PDCs can function to target cancer cells selectively; however, the chemotherapeutic warhead they deliver are often familiar cytotoxic agents. The benefit of conjugating traditional chemotherapeutics to a peptide is that the chemical properties of the cytotoxic agent can be modified. For example, the physiochemical properties of a chemotherapeutic can be changed, such as solubility, selectivity, and half-life [41]. Peptide conjugation can be used on many molecules to create novel PDCs, and the conjugation of peptides with established chemotherapeutic molecules may improve the efficacy of standard therapies.

Typically, the cytotoxic payloads conjugated for PDCs are traditional chemotherapeutic agents, including but not limited to paclitaxel (PTX), camptothecin (CPT), or doxorubicin. Cytotoxic payloads work by various mechanisms of action to arrest mitosis or promote apoptosis in a cancer cell. PTX is a chemotherapeutic used for treating metastatic breast, lung, ovarian, and esophageal cancers that binds to microtubules and induces apoptosis. Despite the relative efficacy of PTX, it is associated with numerous disadvantages, including blood and neurotoxicity, as well as being capable of causing allergic reactions. Furthermore, due to the low solubility of PTX, it must be dissolved in a vehicle, which only adds to the likelihood of developing an allergic reaction. However, solubility was significantly increased without needing a vehicle when conjugated to a cleavable disulfide linker and an octaarginine peptide [42]. The peptide conjugation increased the solubility of PTX without adding the risk of allergic reaction from the vehicle. Doxorubicin is a common chemotherapeutic agent used in many cancer types, including liver, osteosarcoma, and breast [43]. Doxorubicin works as a topoisomerase II inhibitor; however, its physiochemical properties make it incredibly challenging to increase intracellular doxorubicin to therapeutic levels [44]. Modifying doxorubicin into a PDC may improve intracellular concentration and lower associated drug resistance by preventing tumor cells from transporting doxorubicin out of a cancer cell [45].

In addition to classic chemotherapeutic payloads, PDCs can be exploited to deliver short interfering RNA (siRNA) to tumor cells. Once targeted by a CPP, siRNAs can inhibit translation and subsequent protein synthesis. This approach was taken against glioblastoma by He et al., and found that peptide-conjugated siRNA was able to enter the glioblastoma, whereas the unconjugated siRNA was unable and was also able to silence EGFR [46].

Radionuclides are radioactive isotopes that emit particles for either diagnostic or therapeutic uses. Lutetium-177 (^177^Lu) was initially used for treating multiple myeloma but was unsuccessful at increasing long-term survival due to the intravenous mode of administration, low doses, and lower energy β^−^ particles [47]. ^177^Lu conjugated to a peptide allows for sufficient radiation to cells at the site of interest with rapid clearance of radioactivity from non-target tissues and organs [48]. For example, in the phase 3 Neuroendocrine Tumors Therapy (NETTER-1) trial, the estimated risk of death was 60% lower with ^177^Lu-DOTA-TATE versus octreotide LAR [49].

## 5. Lu-177 Current Clinical Application and Trials

Currently, there are 96 clinical trials and 1 FDA approved PDC drug, Lu-177 DOTA-TATE, all targeting overexpressed antigens in solid tumors. PDC clinical trials are in Phases I and II, primarily focusing on the safety and efficacy of the drugs on a limited number of patients.

### 5.1. FDA Approved PDC: Lu-177 DOTA-TATE

The first FDA approved PDC was Lu-177 DOTA-TATE (Lutatera^®^) (Figure 2). Lu-177 DOTA-TATE is a radiolabeled somatostatin analog that was FDA approved in 2018 as a first-in-class drug for the treatment of somatostatin receptor-positive gastroenteropancreatic neuroendocrine tumors (GEP-NETs) and is administered intravenously (I.V.) to patients [16]. NETs are a type of tumor that originate in endocrine tissues throughout the body. Lu-177 DOTA-TATE binds to malignant cells overexpressing somatostatin receptor type 2. Once Lu-177 DOTA-TATE binds its respective target, Lu-177 DOTA-TATE accumulates within tumor cells and delivers cytotoxic radiation to kill the cells [50]. The 3D structures of the somatostatin receptor in complex with somatostatin and octreotide, a synthetic long-acting cyclic octapeptide somatostatin analog, were recently determined by cryo-electron microscopy (Figure 5) [51]. This work has provided a structural basis for the conserved ligand tetrapeptide Phe-Trp-Lys-Thr (FWKT) sequence as a key pharmacophore for receptor binding. This sequence, which is found in several natural somatostatin receptor agonists including somatostatin-14, somatostatin-28 and cortistatin-14, is largely preserved in the chemical structures of octreotide and Lu-177 DOTA-TATE (Figure 5).

### 5.2. Examples of PDC Clinical Trials Utilziing Lu-177

Following the success of ^177^Lu-DOTA-TATE for the treatment of adults with somatostatin receptor–positive GEP-NET and FDA approval, our article focuses on other relevant examples of actively recruiting clinical trials that utilize Lu-177 for treatment of various cancers (Table 1). We identified 44 clinical trials that employ Lu-177 as the payload for PDC cancer therapy.

### 5.3. PDC Limitations

Despite the many advantages of PDCs, there are several limitations to implementing PDC therapy. Due to their low molecular weight, PDCs exhibit poor stability and undergo rapid renal clearance [53,54,55]. This can lead to limited therapeutic utility, particularly on solid tumors. Researchers are working on different peptide chemical and physical modifications to overcome this challenge. An example of a modification technique is using gold nanoparticles (AuNPs) conjugated with PDCs to increase their overall stability. In a study by Kalimuthu et al., PEG-coated-AuNPs were tested to determine if they could provide a suitable platform for loading PDCs [55]. Their research showed that PDCs conjugated with the PEG-coated-AuNPs were still active after a 72 h pre-incubation period. In contrast, the free PDCs had no cytotoxic activity after a 24 h pre-incubation period [55]. Another technique that has been widely used as a method of improving the enzymatic and chemical stability of peptides is the use of cyclization techniques [54]. For example, peptide stapling, a technique that allows peptides to be locked into a desired confirmation, has been used to enhance a peptide’s binding affinity to its target [54].

Another essential factor to consider concerning renal clearance is the overall net charge of the peptide sequence [54]. Increasing the negative charge of the peptide sequence to delay the glomerular filtration by the kidneys has been proposed as a method to lengthen the half-life of peptides [56]. Other ways to delay renal clearance of peptides include increasing the size and hydrodynamic diameter of the peptide and increasing plasma protein binding to prevent the conjugate from being filtered out through the kidneys [56]. One strategy is to conjugate polyethylene glycol (PEG) to PDCs [57]. The inherent qualities of PEG make it an ideal candidate for modification; it is inexpensive, hydrophilic, biocompatible, and non-immunogenic [58]. PDCs possess low immunogenicity compared to ADCs and other more giant molecules, such as proteins. However, they can still benefit from structural modifications to reduce the probability of eliciting an adverse immune reaction. Besides slowing down renal clearance, pegylation can result in less peptide immunogenicity, though PEG has its own immunogenicity issues [59,60], biochemical modification of peptides opens yet another avenue of investigation for beneficial modification of PDCs [54,61,62].

Alternatives to pegylation are also emerging as a way to further modify the biochemical and pharmacokinetic aspects of PDCs, and to reduce the inherent immunogenicity complications of PEG [63]. One such alternative to PEG is polyscarcosine (PSar) which has low toxicity and unlike PEG is biodegradable [64]. Furthermore, the unique chemistry of PSar may mitigate the phenomenon of accelerated blood clearance often observed in pegylated therapeutics [65]. Another alternative to PEG is XTEN a class of unstructured hydrophilic and biodegradable protein polymers that have been demonstrated to increase the half-life of bioactive peptides [66] as well as increasing peptide solubility [67]. A final class of alternatives to PEG include the use of proline/alanine/serine (PAS) biopolymers which can function in many of the same ways a PEG but also share the pharmacokinetic benefits of PEG alternatives such as XTEN and PSar such as increased circulating half-life [68] and decreased immunogenicity [69].

Even after FDA approval, failure to demonstrate a survival advantage presents another critical limitation. A prime example of this is the drug melphalan flufenamide (melflufen), initially approved in February 2021 for the treatment of refractory multiple myeloma; it was withdrawn from the U.S. market in October 2021 after consistently failing to demonstrate a survival advantage based on results from a phase 3 randomized controlled trial [70]. This case highlights the role of the FDA’s regulatory process and accelerated approval pathways, as well as the importance of rigorous clinical trials backed up by robust clinical data before the marketing and approval of a new drug.

PDCs have limited or non-existent oral bioavailability; this limits their administration to intravenous injection and excludes oral administration [54]. Lack of ease of administration presents a tremendous barrier, particularly for individuals who cannot access a clinical setting regularly. Although this is a common challenge for biologics and peptides, further research is needed to optimize their chemical and enzymatic stability. Lamson et al. have recently reported on a novel way of allowing for the oral delivery of bioactive peptides by using anionic nanoparticles [71]. Although using nanoparticles is an attractive strategy to solve the low oral bioavailability and stability problem, and additional testing is needed to evaluate the feasibility of this technique in future clinical trials. The use of acid-stable coatings, gut enzyme inhibitors, and mucus-penetrating peptides have also been proposed as possible strategies to improve the oral availability of PDCs [72]. For example, acid-stable layers can help to slow down the degradation caused by peptidases [72].

Developing approaches to allow for the oral administration of PDCs is essential to make these drugs more accessible, allow for better therapeutic adherence, and to increase their representation in clinical trials. As the number of clinical trials evaluating the use of PDCs grows, further research is needed to optimize their delivery, improve their systemic stability, reduce fast renal clearance, and lengthen their half-life in vivo.

## 6. Conclusions

The need for targeted cancer therapy is more significant than ever. Within the field of targeted cancer therapy, peptides offer unmatched versatility. Established treatments with ADCs have paved the way for the development of PDCs. Compared to other cancer therapy strategies, PDCs offer a more straightforward design, a high degree of selectivity, a broad range of targets, and a wide range of chemical and biological diversity. Although there are several limitations to their implementation, the potential to target tumors and prevent severe adverse effects makes PDCs stand out as attractive alternatives to other cancer therapy methods. The research community is acknowledging the advantages offered by PDCs, and new advances in the field have renewed interest in their use as targeted cancer therapy agents. Several trials are currently underway to determine the efficacy of novel PDCs with non-radioactive payloads and intelligent linkers. Although the development of PDCs has advanced the field of individualized medicine, long-term safety and efficacy are yet to be determined. Similarly, critical technical hurdles to the development of effective PDCs will need to be addressed in the near future.

## Figures and Tables

**Figure 1 ijms-24-00829-f001:**
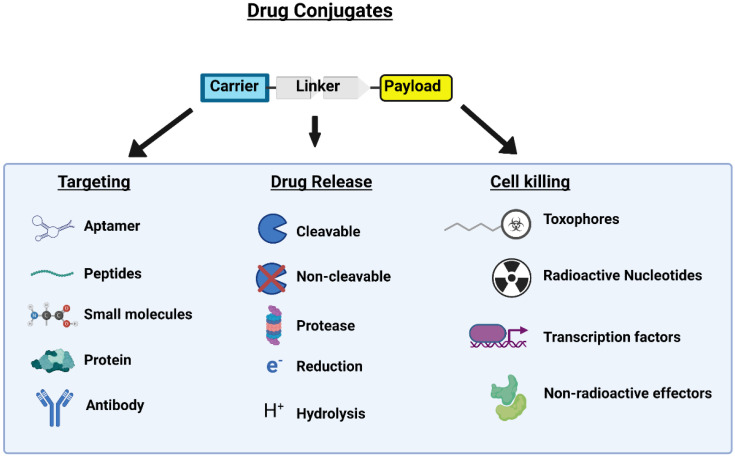
Drug conjugates are composed of 3 parts, carrier, linker and payload. The carrier, also referred to as homing peptides, targets and delivers the drug to the respective tumor target. The payload is the cytotoxin that kills the cancer cells, while the linker region bridges the two PDC components together and induces drug release. This figure was created using BioRender online App and license.

**Figure 2 ijms-24-00829-f002:**
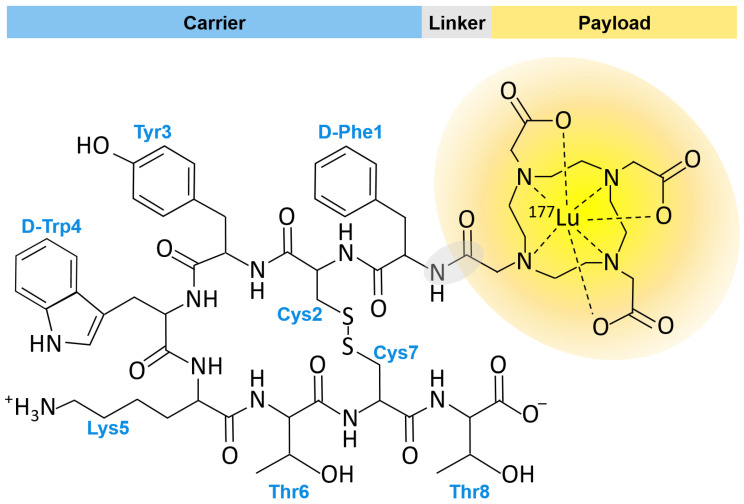
Structure of ^177^Lu-DOTA-TATE (Lutathera^®^ Novartis, Zaragoza Spain). Amino acid residues of the carrier peptide, the tyrosine-containing somatostatin analog Tyr3-octreotate (TATE), are labeled in blue. The payload (highlighted in yellow) is the macrocyclic chelating agent tetraazacyclododecane-tetraacetic acid (DOTA) bound to the beta-emitting radionuclide lutetium-177 (^177^Lu). The linker region (highlighted in grey) consists of an amide bond formed by coupling a carboxyl group of DOTA with the N-terminal amino group of the carrier at D-Phe1.

**Figure 3 ijms-24-00829-f003:**
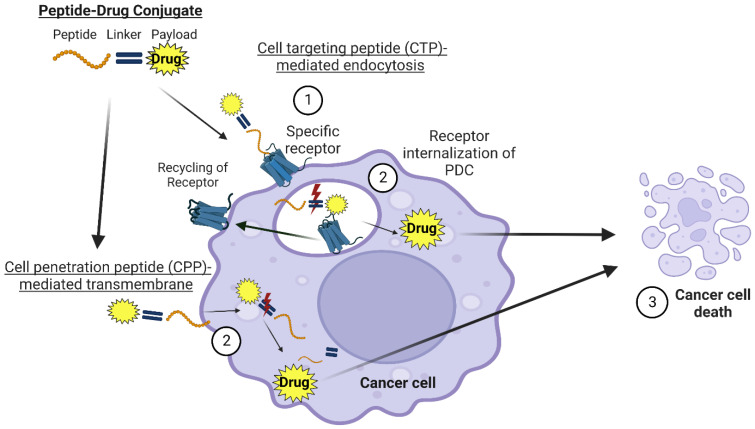
Theoretical summary of the steps that lead to the internalization of the specific (overexpressed receptor) within cancer cells. Step 1 involves the recognition of the PDC by the receptor. Once the PDC is internalized, the drug is released (Step 2). Note: The red lightning bolt depicts the release of the drug. The receptors are separated from the ligand and can be recycled back to the cell membrane. Cleavage of the linker results in drug release and subsequent cancer cell death (Step 3). This figure was created using BioRender software and license.

**Figure 4 ijms-24-00829-f004:**
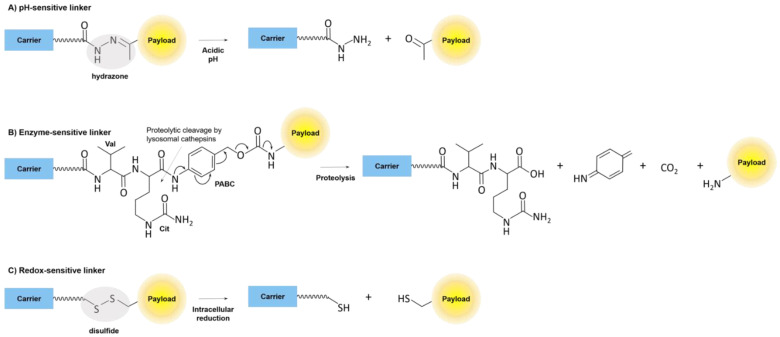
Cleavable linkers. (**A**) A prototypical pH-sensitive linker contains a hydrazone moiety, which is cleaved in an acidic environment such as within an endosome or lysosome. (**B**) An example of an enzyme-sensitive linker is the lysosomal cathepsin-sensitive valine-citrulline-*p*-aminobenzyloxycarbonyl (PABC) function. Chemical rearrangement and release of the PABC moiety during the cleavage reaction enables free release of the payload. (**C**) A redox-sensitive linker is exemplified by a simple disulfide bond, which is reduced by intracellular reducing agents such as glutathione to release the payload.

**Figure 5 ijms-24-00829-f005:**
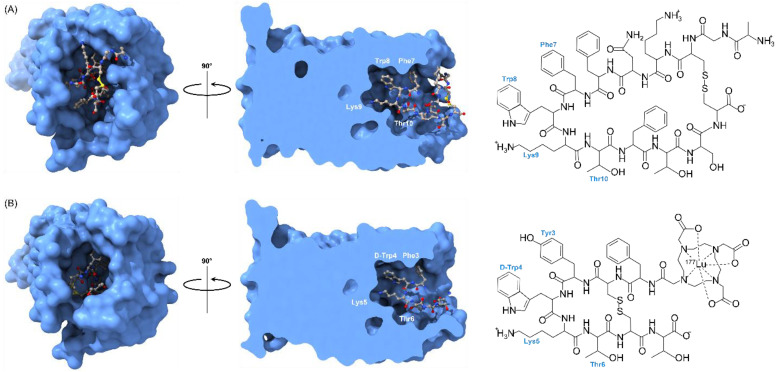
(**A**) 3-dimensional structure of somatostatin-14 (ball-and-stick rendering) in complex with somatostatin receptor 2 (surface rendering) (PDB-ID: 7T10). Residues of the conserved tetrapeptide Phe-Trp-Lys-Thr (FWKT) receptor-binding motif are labeled. The chemical structure of somatostatin-14 is also shown. (**B**) 3-dimensional structure of octreotide (ball-and-stick rendering) in complex with somatostatin receptor 2 (surface rendering) (PDB-ID: 7T11). The chemical structure of ^177^Lu-DOTA-TATE (Luthera^®^), a somatostatin receptor-targeting radiopharmaceutical containing a Tyr3-octreotate carrier function, is also shown. 3-dimensional molecular graphics and analyses performed with UCSF ChimeraX, developed by the Resource for Biocomputing, Visualization, and Informatics at the University of California, San Francisco, with support from National Institutes of Health R01-GM129325 and the Office of Cyber Infrastructure and Computational Biology, National Institute of Allergy and Infectious Diseases [52].

**Table 1 ijms-24-00829-t001:** Current clinical applications and trials of Lutetium-177 (^177^Lu).

Intervention	ClinicalTrials.gov Identifier	Phase	Indication	Target
^177^Lu-PNT2002 versus abiraterone or enzalutamide	NCT04647526	3	Metastatic Castration-resistant Prostate Cancer (mCRPC)	PSMA
^177^Lu-PSMA-I&T versus Hormone Therapy	NCT05204927	3	mCRPC	PSMA
^177^Lu-Ludotadipep	NCT05579184	2	mCRPC	PSMA
^177^Lu-PSMA-617	NCT05114746	2	mCRPC	PSMA
^177^Lu-PSMA (+/−) Ipilimumab and Nivolumab	NCT05150236	2	mCRPC	PSMA
^177^Lu-PSMA and enzalutamide (nonsteroidal antiandrogen)	NCT04419402	2	mCRPC	PSMA
^177^Lu-PSMA (DGUL) and Ga-68-NGUL	NCT05547061	1/2	mCRPC	PSMA
^177^Lu-PSMA-I&T	NCT05383079	1/2	mCRPC	PSMA
Cabazitaxel in combination with ^177^Lu-PSMA-617	NCT05340374	1/2	mCRPC	PSMA
Abemaciclib and 177Lu-PSMA-617	NCT05113537	1/2	mCRPC	PSMA
^177^Lu-rhPSMA-10.1	NCT05413850	1/2	mCRPC	PSMA
^177^Lu-EB-PSMA-617	NCT03780075	1	mCRPC	PSMA
^177^Lu-PSMA-EB-01 (+/−) radioligand therapy (RLT)	NCT05613738	1	mCRPC	PSMA
^177^Lu-PSMA + olaparib (PARP inhibitor)	NCT03874884	1	mCRPC	PSMA
^177^Lu-EB-PSMA (55 mCi)	NCT04996602	1	mCRPC	PSMA
^177^Lu-Ludotadipep	NCT05458544	1	mCRPC	PSMA
^177^Lu-DOTA-TLX591	NCT04786847	1	mCRPC	PSMA
Radiometabolic Therapy (RMT) with ^177^Lu PSMA 617	NCT03454750	2	Castration Resistant Prostate Cancer (CRPC)	PSMA
^177^Lu-PSMA-617	NCT04443062		Oligo-metastatic Hormone Sensitive Prostate Cancer (mHSP)	PSMA
Standard of Care (SOC) (+/−) ^177^Lu-PSMA-617	NCT04720157		mHSPC	PSMA
Docetaxel +/− ^177^Lu-PSMA	NCT04343885	2	metastatic hormone-naive prostate cancer (mHNPC)	PSMA
^177^Lu-TLX591	NCT05146973	2	PSMA-expressing prostate cancer	PSMA
225Ac-J591 and ^177^Lu-PSMA-I&T	NCT04886986	1/2	Prostate cancer	PSMA
^177^Lu-PSMA	NCT05230251	2	Prostate cancer	PSMA
^177^Lu PSMA 617	NCT04663997	2	Prostate cancer	PSMA
^177^-Lu-PSMA given before stereotactic body radiotherapy (SBRT)	NCT04597411	2	Prostate cancer	PSMA
^177^Lu-PSMA-617	NCT05613842	2	Hormone-sensitive disease (cohort A) castrate-resistant Disease (Cohort B)	PSMA
^177^Lu-PSMA radioligand therapy	NCT05162573	1	node-positive prostate cancer	PSMA
^177^Lu-PP-F11N	NCT02088645	1	Advanced medullary thyroid carcinoma GEP-NET	cholecystokinin-2 receptors
^177^Lu-AB-3PRGD2	NCT05013086	1	Non-Small Cell Lung Cancer (NSCLC)	Integrin αvβ3
^177^Lu-DOTA-TATE in combination with carboplatin, etoposide, and tislelizumab	NCT05142696	1	Extensive Stage Small Cell Lung Cancer (ES-SCLC)	STTR
GD2-SADA:177Lu-DOTA complex	NCT05130255	1	GD2 expressing solid tumors (Small Cell Lung Cancer, Sarcoma and Malignant Melanoma)	GD2
Standard of Care (+/−) ^177^Lu-DOTA-TATE	NCT05109728	1	Glioblastoma	STTR
Intracavitary radioimmunotherapy (iRIT) with a newly developed radioimmunoconjugate ^177^Lu labeled 6A10-Fab-fragments	NCT05533242	1	Glioblastoma	carbonic anhydrase XII
Combination of ^177^Lu-girentuximab and nivolumab	NCT05239533	2	Advanced clear cell renal cell carcinoma/ccRCC	Carbonic Anhydrase IX
68Ga-PSMA PET-CT with ^177^Lu-EB-PSMA-617	NCT05170555	NA	Renal Cell Carcinoma	PSMA
^177^Lu-PNT6555	NCT05432193	1	Fibroblast Activation Protein (FAP) overexpressing tumors (Colorectal Cancer; Esophageal Cancer; Melanoma; Soft Tissue Sarcoma	FAP
[68Ga]Ga DOTA-5G and ^177^Lu DOTA-ABM-5G theranostic	NCT04665947	1	Locally advanced or metastatic pancreatic adenocarcinoma (PDAC)	-
^177^Lu-octreotate versus sunitinib	NCT02230176	2	Progressive pancreatic, inoperable, somatostatin receptor positive, well differentiated pancreatic neuroendocrine tumors (WDpNET).	STTR
^177^Lu-DOTATATE versus capecitabine and temozolomide	NCT05247905	2	Metastatic Pancreatic Neuroendocrine Tumor and Unresectable Pancreatic Neuroendocrine Carcinoma	STTR
^177^Lu-DOTATATE hepatic intraarterial infusion	NCT04544098	1	Neuroendocrine Tumors Liver-Dominant Metastatic Pancreatic Neuroendocrine Tumors	STTR
^177^Lu-DOTATOC	NCT04276597	2	Somatostatin receptor-expressing Pulmonary, Pheochromocytoma, Paraganglioma, and Thymus neuroendocrine tumors	STTR

## Data Availability

Not applicable.

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
