# Peer review of "Peptide Drug Conjugates and Their Role in Cancer Therapy"

_ijms, 2023, doi:10.3390/ijms24010829_

Round 1

Reviewer 1 Report

The authors summarize the application of peptide-drug conjugates, whose peptide parts were broken down to CPP and CTP. Overall the article is interesting, well-written, and worthy of publishing. However, the reviewer feels that a little amendment may strengthen the overall claim. In particular, a clear comparison between ADC and PDC should be considered. 

  1. I believe that one of the advancements of PDC is rapid internalization compared with ADCs. If the authors summarize the comparison of the internalization rate of both conjugates (ADC: XX% and PDC: YY%), it should strengthen this manuscript. Of course, these rates depend on the behavior of the binder molecule (mAb VS peptide), but rough prediction still is useful
  2. One remaining challenge of PDC is the risk of immunogenicity. The authors should have comments about this topic. To summarize current development of immunogenicity may be useful.
  3. A summary of organ approaches would be useful. For example, drug delivery to the lungs is relatively easy to achieve with GalNAc ligands, but drug delivery to the brain and pancreas is still difficult.
  4. The reviewer agrees with the advantage of the use of PEG moiety to increase the half-life of PDC, but PAS (Proline-Alanine-Serine), Polysarcosine, Xten, or other PEG alternative is also useful. Authors may add comments about these molecules.

Author Response

Reviewer #1.

Reviewer Comment:

The authors summarize the application of peptide-drug conjugates, whose peptide parts were broken down to CPP and CTP. Overall the article is interesting, well-written, and worthy of publishing. However, the reviewer feels that a little amendment may strengthen the overall claim. In particular, a clear comparison between ADC and PDC should be considered.

Authors’ Response:

We thank the reviewer for this feedback on our article. A comparison between ADC and PDC has been added to Section1, lines 68-90. Added comments/changes in text are in red.

Reviewer Comment:

I believe that one of the advancements of PDC is rapid internalization compared with ADCs. If the authors summarize the comparison of the internalization rate of both conjugates (ADC: XX% and PDC: YY%), it should strengthen this manuscript. Of course, these rates depend on the behavior of the binder molecule (mAb VS peptide), but rough prediction still is useful.

Authors’ Response:

The internalization rate of both conjugates ADC and PDC has been added to Section 1,  lines 79-84. Added comments/changes in text are in red.

Reviewer Comment:

One remaining challenge of PDC is the risk of immunogenicity. The authors should have comments about this topic. To summarize current development of immunogenicity may be useful.

Authors’ Response:

In addition to the internalization rate of ADCs and PDCs, the comparison of the risk of immunogenicity has been added to Section 1, lines 70-78. Added comments/changes in text are in red.

Reviewer Comment:

A summary of organ approaches would be useful. For example, drug delivery to the lungs is relatively easy to achieve with GalNAc ligands, but drug delivery to the brain and pancreas is still difficult.

Authors’ Response:

We appreciate the authors suggestion to include a discussion of how PDCs may be used to target specific organs we have added a section (2.1) on organ targeting lines 134-148. Added comments/changes in text are in red.

Reviewer Comment:

The reviewer agrees with the advantage of the use of PEG moiety to increase the half-life of PDC, but PAS (Proline-Alanine-Serine), Polysarcosine, Xten, or other PEG alternative is also useful. Authors may add comments about these molecules.

Authors’ Response:

We agree that the use of PEG alternatives is an important element to be discussed. We have added additional discussion of PAS, XTEN, and Polysarcosine in the limitations section 6.3 lines 397-398; 401-412. Added comments/changes in text are in red.

Reviewer 2 Report

See attachment

Author Response

Reviewer 2:

Reviewer Comment:

This mini-review covers many aspects of PDC and is informative to readers. However, there are several review articles of PDCs have been published including an in-press comprehensive article published by Fu et al. in Acta Pharmaceutica Sinica B, 2022, https://doi.org/10.1016/j.apsb.2022.07.020. The authors need to cite those articles and state the difference between this manuscript and other published reviews. Does it include more recent technology advancement or give an update on recent development? Other comments are showing below. The manuscript is recommended for publication with major revision.

Authors’ Response:

We appreciate the reviewer’s comments and recommendations and have added the comprehensive article published by Fu et al. in Acta Pharmaceutica Sinica B, 2022 to our article and reference list.

We have also modified our article’s innovative aspect to focus primarily on Lutetium Lu 177, as Lutetium Lu 177 dotatate (Lutathera®), is the only FDA approved PDC. We also have a section that includes FDA information on Lutetium Lu 177 dotatate (Lutathera®).

Our review focuses more on CTP and CPP. Another innovative aspoect of our article is the utilization of emerging structural biology (3D structure of drug complex). Added comments/changes in text are in red.

Reviewer Comment:

P1, line 53, add a comma after women.

Authors’ Response:

We have added a comma after women in the sentence “Cancer is a significant cause of death worldwide, with breast and lung cancer having the highest prevalence among women, and lung and prostate cancer having the highest prevalence among men.” section 1 line 33. Added comments/changes in text are in red.

Reviewer Comment:

P4, line 105, it should be “The peptides in PDCs are classified as cell-penetrating peptides (CPPs) or cell-targeting peptides (CTPs)”, not PDCs. In addition, published review articles of CPPs and CTPs need to be cited.

Authors’ Response:

The sentence in section 2, lines 130-131 has been modified to read “The peptides in PDCs are classified as cell-penetrating peptides (CPPs) or cell-targeting peptides (CTPs).” Added comments/changes in text are in red. New citations were added. 

Reviewer Comment:

P5, line 166, it will be good to give the 18-amino acid sequence of pVEC.

Authors’ Response:

We agree with this feedback and have added the 18-amino acid sequence of pVEC to section 2.3 line 210.

Reviewer Comment:

P5, the authors report four types of linkers for PDCs. It will be good to have a figure to show the linkage of various linkers, instead of just brief description.

Authors’ Response:

In addition to the brief description of the types of linkers, Figure 4 has been added to our article. Added comments/changes in text are in red

Reviewer Comment:

P5, in section 3.3. Enzyme-sensitive linkers, the reviewer believes there is a typo of ester (not ether) throughout this section. The ether bind can not be cleaved by esterases.

Authors’ Response:

We thank the reviewer for identifying this typo. All ether references have been changed to ‘ester’ in Section 3. Added comments/changes in text are in red

Reviewer Comment:

P8, the authors report 5 PDCs including a vaccine in clinical trial. It will be good to describe the peptides (e.g., sequence) in those PDCs. There are many more PDCs in clinical trial (see Acta Pharmaceutica Sinica B, 2022, https://doi.org/10.1016/j.apsb.2022.07.020). Why the authors only selected five?

Authors’ Response:

We appreciate this crucial feedback from the reviewer and have substantially modified Table 1. Our focus is on the most prescient clinical trials as candidates for newly emerging therapeutics. Table 1 now includes an extensive list of actively recruiting clinical trials that primarily focused on lutetium Lu 177 as the payload. Our review focuses on clinical trials in the active recruitment phase. The clinical trials that were completed, terminated, or not yet recruiting were not included in our table. Added comments/changes in text are in red.
